Biological characteristics of marine Streptomyces SK3 and optimization of cultivation conditions for production of compounds against Vibiriosis pathogen isolated from cultured white shrimp (Litopenaeus vannamei)

Khaochamnan Rachow 1
Suanyuk Naraid 1
http://orcid.org/0000-0001-7777-8062 Lertcanawanichakul Monthon 2
Pedpradab Patchara 3 ppedpradab@gmail.com
1 Department of Aquatic Sciences, Faculty of Natural Resources, Prince of Songkla University , Hat-Yai, Songkhla , Thailand
2 School of Allied Health Sciences, Walailak University , Thasala, Nakhonsithammarat , Thailand
3 Department of Marine Science, Faculty of Sciences and Fishery Technology, Rajamangala University of Technology Srivijaya , Sikao, Trang , Thailand
Beddoe Travis
Electronic publication date: 2024 Sep 24
Publication date: 2024
Volume: 12
Electronic Location ID: e18053
Received 2024 Jan 23; Accepted 2024 Aug 16
Copyright: © 2024 Khaochamnan et al.
Copyright year: 2024
Copyright holder: Khaochamnan et al.
License: This is an open access article distributed under the terms of the Creative Commons Attribution License, which permits unrestricted use, distribution, reproduction and adaptation in any medium and for any purpose provided that it is properly attributed. For attribution, the original author(s), title, publication source (PeerJ) and either DOI or URL of the article must be cited.
License URL: https://creativecommons.org/licenses/by/4.0/

Keywords: Anti-vibrio spp, Marine streptomyces, Culture, Characterization, Optimization, Litopenaeus vannamei

Funding: Thailand Science Research and Innovation through RUTS This work is supported by Thailand Science Research and Innovation through RUTS. The funders had no role in study design, data collection and analysis, decision to publish, or preparation of the manuscript.

==============================
Antibiotic resistance in shrimp farms has emerged as an extremely serious situation worldwide. The main aim of this study was to optimize the cultural conditions for producing new antibiotic agents from marine Streptomyces species. Streptomyces SK3 was isolated from marine sediment and was identified by its 16S rDNA as well as biochemical characteristics. This microbe produced the highest concentration of bioactive secondary metabolites (BSMs) when cultured in YM medium (YM/2). It produced the maximum total protein (41.8 ± 6.36 mg/ml) during the late lag phase period. The optimum incubation temperature was recorded at 30 °C; BSMs were not produced at ≤10 °C within an incubation period of 3–4 days. The suitable agitation speed was found to be 200 rpm with pH 7.00. The proper carbon, nitrogen, and trace elements supplementation consisted of starch, malt extract, calcium carbonate (CaCO3), and magnesium sulfate (MgSO4). The ethyl acetate extract was found to act strongly against three vibriosis pathogens, Vibrio harveyi, Vibrio parahaemolyticus, and Vibrio vunificus, as indicated by the inhibition zones at 34.5, 35.4, and 34.3 mm, respectively. The extract showed the strongest anti-V. harveyi activity, as indicated by minimum inhibitory concentration (MIC) and minimum bactericidal concentration (MBC) values of 0.101 ± 0.02 and 0.610 ± 0.04 mg/ml, respectively. Basic chemical investigation of the crude extract using thin layer chromatography (TLC), bioautography, liquid chromatography tandem mass spectrometry (LC‒MS/MS), Fourier transform infrared spectroscopy (FTIR), and proton nuclear magnetic resonance (1H-NMR) revealed that the active components were the terpenoid and steroid groups of compounds. They showed carboxylic acid and ester functions in their molecules.

Introduction

Antimicrobial resistance (AMR) is a global long-term crisis. The World Health Organization (WHO) considers it one of the most critical problems facing humanity, as it is estimated that AMR could cause the death of 10 million people worldwide in 2050 (Caneschi et al., 2023). The current AMR situation is not a challenge to prevent but is also aggravated by the fact that novel antibiotics or lead molecules will not be developed and launched in the next decade (Singh, Mazumder & Bora, 2009; Vercelli et al., 2022). Furthermore, many reports have confirmed that AMR can transfer from animals to humans (Okeke et al., 2022). The AMR phenomenon not only occurs in the human community but also widely spreads to livestock environments, particularly in the aquaculture sector. In the shrimp culture environment, bacteria are the major pathogenic infection, with an emphasis on the high mortality Vibrio species that cause acute hepatopancreatic necrosis disease (AHPN) (Schar et al., 2021).

The AMR burden in shrimp culture has been increasing due to farmers using antibiotics for both treatment and prophylaxis, and there is a lack of alternative antibiotic choices to prevent crop failure (Thornber et al., 2020). To address the severity of the AMR situation in the next decade, there is an urgent need to focus on the discovery of novel antimicrobial metabolites derived from natural resources. This because the new antibiotics will be discovered and their structure diversity will also be evaluated. In this context, Actinobacteria is a promising source of these metabolites. Actinomycetes are gram-positive bacteria with a filamentous shape and branching structure similar to fungi. They thrive in diverse environments, both marine and terrestrial, and have tremendous potential for producing a wide range of novel bioactive secondary metabolites (BSMs). The synthesis of BSMs is controlled by genetic materials and is regulated in response to environment (Jagannathan et al., 2021). Among several genera, Streptomyces is the largest member with the richest source of antimicrobial metabolites. They are gram-positive aerobic, saprophyte bacteria with high G+C content in DNA of approximately 69–78% and have multiple biosynthesis-related gene clusters (BGCs) on each genome that involve diverse BSMs synthesis (Alam et al., 2022). Approximately 70–80% of antibiotics currently have been isolated from Streptomyces (Alam et al., 2022). Streptomyces synthesize BSMs during physiological and morphological differentiation, which involve stress (both caused by nutrient deficiency and undesired environmental factors) and biosynthesis activation. In critical and complex ecosystems, such as high salt content and temperature and acidic and alkaline environments, these inappropriate environments can stimulate Streptomyces to synthesize novel BSMs, which are released outside the cells to serve biological functions such as self-defense, communication signals and allelopathic proposes (Croce et al., 2021). The BSM synthesis process begins with slow and declining growth; therefore, it generally occurs in the stationary phase, and maximum yield production is observed during the decline phase. In cultural systems, BSM synthesis is greatly influenced by the optimization of fermentation conditions, including nitrogen, carbon and phosphorus sources, mineral salts, metal ions, types of precursors, inducers and inhibitors, which vary among species (Subramani & Aalbersberg, 2012). The identification and characterization of Streptomyces is definitely complicated and challenging as a result of a large number of described species having relative similarity to others within the genus and still unclear descriptions. To clarify the identification and characterization, several criteria may need to be integrated. Selective media are suggested for isolating Streptomyces from natural resources, while identification and characterization are primarily performed by observing their morphological characteristics, such as spore chain, color of substrate and aerial mycelia, when cultured in different media as a standard method proposed by the International Streptomyces Project (ISP) (Law et al., 2018). To date, 16S rDNA gene sequencing is the most powerful tool used to identify the isolated Streptomyces; however, due to the very large size in this genus, it does not precisely identify when high similarity gene sequencing has discovered and unclear phylogenetic relationships. In this case, additional methods are used to integrate the identification, including physiological and biochemical examinations.

Despite terrestrial streptomycetes being a major source of antibacterial discovery, marine streptomycetes are also reported as a repository source of novel antibacterial compounds (Donald et al., 2022). Currently, the discovery of antibiotic metabolites from marine Streptomyces is still limited, with approximately 20 compounds possessing very high potential for antibacterial activity having been isolated since 2000 (Manivasagan et al., 2014). Moreover, only 10% of Streptomyces are isolated from marine environments due to limited recognition of isolation and cultural conditions. Ordinary cultivation yields fewer strains, resulting in physical, chemical and structural distinctions in comparison to terrestrial habitats (Alenazi et al., 2023). In marine environments, Streptomyces survive among critical, stress and dynamic environments, such as high pressure, fluctuating temperature, high salt content, various substrates, and nutrient variation. These factors induce Streptomyces to adapt both biologically and chemically by producing specific secondary metabolites, particularly those with antimicrobial activity. To explore marine actinomyces, researchers have to develop spatial isolation techniques, media composition and cultural optimization to cover a wide range of genera for large-scale culturing because different isolated sources may need different media and culture conditions (Jose, Maharshi & Jha, 2021).

In this study, we present the isolation and characterization of a marine Streptomyces SK3 strain. We examined the potential anti-vibriosis pathogens isolated from white shrimp (Litopenaeus vannamei) collected in a vibriosis pandemic area.

Materials and Methods

The methods subsequently covered four main parts including biological characterization of Streptomyces SK3, optimization of cultivation, isolation of tested organisms and biological screening, and chemical screening part as describe following.

Biological characterization

Isolation and purification of bacterial samples

The bacterial sample was isolated from marine sediment in the intertidal zone (Gao et al., 2022) of Sarai Island (6°39′.97″ N, 99°51′.32.01″ E), Satun Province, Thailand, in 2022. A sediment sample was collected using a core sampler, transferred to a sterile plastic bag, and transported to the laboratory in an icebox. One gram of sediment was dispersed in 1 ml of sterile artificial sea water. After 5 min of mixing on a shaker, the bacterial suspension was serially diluted with artificial sea water (101–10−5 dilutions). One milliliter of each dilution was spread on a Zobell marine agar plate (HiMedia Laboratories, Thane, India). The plates were placed in the incubation chamber at a temperature of 30 °C for 7 days and observed on a daily basis. The apparent colonies were picked out by needle and re-streaked on Zobell marine medium plates to obtain pure strains, which were preserved in the same media slant at 4 °C for further experiments. For long-term preservation, 15% glycerol was added before the strain was frozen at −80 °C.

Identification and characterization of the isolated bacteria

Biochemical characteristics

Biochemical characteristics were used to identify and confirm the species level of the isolated strain, according to the method described by Chang et al. (2008) and MacFaddin (2000). Two types of biochemical properties of the isolated bacteria were examined: utilization of carbon sources and substrate hydrolysis. A variety of sugars, such as arabinose, fructose, and glucose, were used to characterize the utilization of carbon sources. Substrate hydrolysis was performed using adenine, casein, esculin, gelatin, hypoxanthine, keratin, nitrate, tyrosine, urea, xanthine, and xylan as testing materials.

16S rDNA gene amplification and sequencing

For 16S rDNA identification, DNA templates for polymerase chain reaction (PCR) amplification were prepared using the Genomic DNA Mini Kit (blood/culture cell) (Geneaid Biotech Ltd., New Taipei, Taiwan). DNA coding for 16S rRNA regions was amplified through PCR with Taq polymerase, as described Katsura et al. (2001), Kawasaki et al. (1993), and Yamada et al. (2000). A PCR product was prepared by using the following two primers: 20F (5′-GAG TTT GAT CCT GGC TCA G-3′, positions 9–27 on 16S rDNA) and 1500R (5′-GTT ACC TTG TTA CGA CTT-3′, position 1509-1492 on 16S rDNA) as per the E. coli numbering system (Kawasaki et al., 2008; Yamada et al., 2000; Brosius et al., 1981). PCR amplification performed by following the method described by Lertcanawanichakul & Chawawisit (2019). Briefly, BIO RAD DNA Engine Dyad Peltier Thermal Cycler PTC-220 (Bio-Rad Laboratories, Hercules, CA, USA) was used as amplification instrument. The mixture of 15–20 ng of template DNA, 2.0 µmoles of two designed primers, 2.5 units of Taq polymerase, 2.0 mM of magnesium chloride, 0.2 mM of deoxynucleoside triphosphate (dNTP), 10 µl of 10xTaq buffer at pH 8.8 and ammonium sulfate were used as reaction solution. The PCR amplification program followed to the method described by Lertcanawanichakul & Chawawisit (2019). The product of PCR reaction was examined on 0.8% (w/v) agarose gel electrophoresis and the target DNA was further purified by standard PCR Kit (Geneaid Biotech Ltd., New Taipei City, Taiwan). Direct sequencing of the single-banded and purified PCR products (ca. 1,500 bases on 16S rDNA by the E. coli numbering system) (Moran, Rutherford & Hodson, 1995) was carried out. Next, sequencing of the purified PCR products was performed on an ABI Prism® 3730XL DNA Sequence (Applied Biosystems, Foster City, CA, USA) by sequencing the service provider. Two primers, 27F (5′-AGA GTT TGA TCM TGG CTC AG-3′) or 800R (5′-TAC CAG GGT ATC TAA TCC-3′), and 518F (5′-CCA GCA GCC GCG GTA ATA CG-3′) or 1492R (5′-TAC GGY TAC CTT GTT ACG ACT T-3′), were used for single strand 16S rDNA sequencing. Four primers, 27F, 518F, 800R, and 1492R, were used for double-strand 16S rDNA sequencing. For sequence analyses, the nucleotide sequences obtained from all the primers were assembled using the cap contig assembly program, an accessory application in the biological sequence alignment editor (BioEdit) program (https://bioedit.software.informer.com/). The identification of phylogenetic neighbors was initially carried out using the nucleotide basic local alignment search tool (BLASTn) (Altschul et al., 1997) program against a 16S rDNA sequence database of validly published prokaryotes. The sequences with the highest scores were calculated to obtain pairwise sequence similarity using a global alignment algorithm (Myers & Miller, 1998).

Scanning electron microscopy

The specimen was prepared according to the method described by MacFaddin (2000). Briefly, the microbe in liquid medium was filled with filter paper; next, the filter paper containing Streptomyces cells was dried on stubs and then coated with a gold/platinum mixture for 75 s to achieve a final thickness of 180 nm. The micrograph was obtained using a field emission scanning electron microscope (SEM) (JSM-IT800; JEOL, Ltd, Schottky, CA, USA).

Optimization of cultivation (laboratory based methods)

Culture media optimization

To assess the types of potential media and culture conditions used for the mass production of BSMs, the isolated Streptomyces were initially cultured in nine different media (international Streptomyces project (ISP) No. 3, 4, 5 and 7), Yeast malt broth (YM), Tryptic soy broth (TSB), Mueller hinton broth (MHB), and Lysogeny broth (LB) at pH 7 and a temperature of 30 °C for 5 days, according to the method previously described (Bawazir, Shivanna & Shantaram, 2018). Streptomyces cultured in liquid media were screened for anti-vibriosis pathogens using the agar well method, and growth performance was determined on a daily basis during the cultivation period.

Media optimization for mass production of BSMs

The optimization of culture media was performed according to method described by Al-Ansari et al. (2020) by reduced concentrations of standard media calculated as 1/2, 1/3, 1/4, 1/5, and 1/6 parts of normal use. Streptomyces cultured in modified media were cultured under the same conditions as previously described; they were screened for anti-vibriosis potency daily. The selected media were examined for BSM production factors, including incubation periods, agitation pH, carbon and nitrogen source utilization, salt content, and trace elements, by modifying the method described by Schrader & Blevins (2001).

Mass culture and extraction of bioactive secondary metabolites

Large scale production performed by modified method described by Bundale et al. (2015). Pure isolated bacterium was picked to 5 ml of culture broth (HiMedia Laboratories, Thane, India) and incubated at 30 °C on a rotary shaker (200 rpm) for 5 days. It was then transferred to 250 ml conical flasks containing 180 ml of broth medium, then cultured under the same conditions as described above; cell density was determined daily. The bacterial suspension at a density of 108 cell/ml was transferred to 1,000 ml baffled flasks containing 500 ml of selected liquid medium to be cultured for 5 days (late lag phase) before harvesting. The cell suspension was placed in an ultrasonic bath at a temperature of 30 °C for 5 min. Subsequently, it was extracted three times with ethyl acetate in a separatory funnel. This process resulted in two distinct liquid layers: one containing the organic layer (ethyl acetate) and the other containing the aqueous part (broth). The organic layer was then evaporated using a rotary evaporator at a water temperature of 45 °C, yielding the crude extract.

Isolation of tested organisms and biological screening

Isolation of Vibriosis pathogens

Three vibriosis pathogens, V. harveyi, V. parahaemolyticus, and V. vulnificus, were isolated from Pacific white shrimp (Litopenaeus vannamei) in the pandemic-affected area of Songkhla Province, Thailand by Asst. Prof. Dr. Naraid Suanyuk at Kidchakan Supamattaya Aquatic Animals Health Research Center (KSAAHRC). The pathogens were isolated and purified by using the 16S rDNA method previously described by Khimmakthong & Sukkarun (2017). The pure strains (V. Harveyi PSU-KSAAHRC-4, V. Parahaemolyticus PSU-KSAAHRC-24, and an un-coded V. Vulnificus) have been deposited at KSAAHRC.

Primary screening of anti-Vibriosis pathogens

Agar-well diffusion methods (Fahmy, 2020) were employed to screen for anti-vibriosis activity (V. harveyi, V. vulnificus and V. parahaemolyticus). The test solution was prepared by dissolving 10 mg of a crude extract in a 1 ml solute containing 2% dimethyl sulfoxide (DMSO) in distilled water (v/v). A bacterial suspension was prepared at a cell density equal to McFarland No. 0.5 in distilled water before being spread in a liquid medium. After drying, the 6 mm diameters of the wells were aseptically punched using a sterile borer. Fifty microliters of the testing solution was transferred to the wells before being incubated at 30 °C for 24 h to measure the inhibition zone (Puangpee & Sanyuk, 2021). Two percent DMSO and tetracycline drug were used as negative and positive controls, respectively.

Determination of minimum inhibitory concentration

The minimum inhibitory concentration (MIC) was determined using a broth dilution method employing a 96-well microtiter plate according to the method described by Puangpee & Sanyuk (2021). The tested extracts were dissolved in DMSO as described previously and diluted serially to concentrations of 10−0.15 mg/ml. A total of 100 μl of test strain was transferred to a well, followed by the addition of 50 μl of the extract. A plate was covered and incubated at 30 °C for 24 h. Tetracycline and 2% DMSO in water were used as positive and negative controls, respectively. After incubation, the plate was detected using a microplate reader (BMG Labtech, Ortenberg, Germany) at a wavelength of 690 nm. MIC values were recorded at the lowest concentration inhibiting vibriosis growth. The minimum bactericidal concentration (MBC) was determined by using 50 μL dispensed in all wells with no visible bacterial growth to streak on MHA and was incubated. The lowest concentration that inhibited 99.9% of the tested organisms was recorded as the MBC value.

Chemical screening

Directed thin layer chromatography bioautography analysis

Thin layer chromatography (TLC)-directed bioautography analysis was used to observe the position of active metabolites as previously described by Khawchamnan et al. (2021).

Chemical analysis of extract

Chemical analysis of crude extracts was carried out using TLC and liquid chromatography tandem mass spectrometry (LC‒MS/MS) by following the method described by Rateb et al. (2011). The silica gel F254 TLC glass sheet (Sigma‒Aldrich, Damstat, Germany) was cut into appropriate sizes before spotting it with the extract; it was then developed in a mobile phase, allowed to dry, and dipped into a tested suspension (106 CFU/ml). The TLC bioautogram was incubated at 25 °C for 48 h under humid conditions and further visualized by spraying with tetrazolium salt. The clear white zone against a purple background indicates the position of active constituents. LC‒MS/MS measurements were carried out using a liquid chromatography-quadrupole time-of-flight mass spectrometer (LC-QTOFMS), A1290 Infinity (Agilent Technologies, Santa Clara, CA, USA). A Zorbax SB-C18 was used as a separation column (Rapid Resolution HD, 2.1 × 150 mm, 1.8 µm; Agilent Technologies, Santa Clara, CA, USA) with an injection volume of 10 µl. The LC system was connected to an Agilent Technologies 6490 Triple Quad electrospray ionization mass spectrometer (ESIMS; Agilent Technologies, Santa Clara, CA, USA). Both negative and positive modes were detected.

Statistical analysis

All of measurement data were calculated by using spss program (version 22). One-way ANOVA analysis was performed and the mean values were analysis for their statistical different at 95% (p = 0.05) confidence interval using Duncant New Multiple Rang Test (DMRT).

Results

The isolated bacteria, SK3, cultured on MA medium was found to have a characteristic orange‒yellowish circular shape with oily smooth skin embedded in solid media. The cell features viewed under an SEM showed a long branching shape (Fig. 1). The 16S rDNA analysis revealed that the nucleotide sequence was compatible with S. hiroshimensis, having a similarity of 99.40%. The phylogenetic analysis (Fig. 2) found that the evolutionary hierarchy branched out from S. hiroshimensis. Therefore, SK3 was further examined for its biochemical and physiological characteristics, as shown in Tables 1 and 2, respectively.

Figure 1 Colony feature of S. hiroshimensis (SK3 strain) and SEM image.

Photographed by Patchara Pedpradab, 8 February, 2024.

Figure 2 Phylogram of phylogenetic relation based on 16S rDNA sequence of S. hiroshimensis.

Table 1 Biochemical characteristic of SK3 comparison to S. hiroshimensis (BCRC1243).

Characteristic	SK3	S. hiroshimensis (BCRC1243, Ref)	
Growth temperature (°C)	25–37	15–40	
Melanin production	+	+	
Lysozyme resistance	–	–	
Carbon utilization	–	–	
Arabinose	–	–	
Fructose	–	–	
Cellulose	–	–	
Glucose	+	+	
Inositol	+	+	
Mannitol	–	–	
Raffinose	–	–	
Salicin	–	–	
Starch	–	–	
Sucrose	–	–	
Xylose	–	–	
Substrate hydrolysis			
Adenine	–	–	
Casein	+	+	
Esculin	–	–	
Gelatin	–	–	
Hypoxanthine	+	+	
Keratin	–	–	
Nitrate	+	+	
Tyrosine	+	+	
Urea	–	–	
Xanthine	–	–	
Xylan	–	–	

Table 2 Physiological characteristics of S. hiroshimensis.

Characteristics	Results	
Gram staining	Positive	
Shape and growth	Filamentous aerial growth	
Production of diffusible pigment	+	
Medium growth	YM/2 and ISP No. 3	
Growth temperature	25–37 °C	
Optimum growth temperature	30 °C	
Range of pH for growth	4–12	
Optimum pH	7	
Salinity	5–40 ppt	
Salt content (NaCl)	0.5%	
Note:

+, possess; YM/2, Yeast Malt broth (half formula); ISP 3, ISP Medium No. 3.

Antivibriosis species screening

An ethyl acetate extract was prepared from S. hiroshimensis (SK3 strain) and screened for anti-vibriosis activity (V. vulnificus, V. harveyi, and V. parahaemolyticus). The results showed that the extract and supernatant strongly inhibited all the tested microorganisms (Fig. 3 and Table 3). Bioautogram and TLC patterns of the extract revealed the existence of a diverse group of active components (Fig. 4). The component inhibiting V. harveyi reacted with both vanillin and anisaldehyde to produce a green band, whereas the component active against V. Vulnificus produced gray and pink bands upon reacting with vanillin and anisaldehyde. Finally, the compounds inhibiting V. parahaemolyticus reacted with vanillin and anisaldehyde, producing purple bands on the TLC sheet.

Figure 3 Inhibition zone of anti-Vibrio spp. screening of the extract from S. hiroshimensis.

(I) and media optimization (II) determined by agar well diffusion method. (A) Tetracycline antibiotic, (B) supernatant, (C) crud extract, (D) pure media, For media optimization process, the supernatant of suspension cell in the media was screened for their anti-V. harveyi property; (E) YM/2, (F) YM, (H) ISP3, (G) ISP3/2. Photographed by Patchara Pedpradab, 1 February, 2024.

Table 3 MIC and MBC of the extract from strain SK3 against Vibriosis pathogen.

Vibriosis pathogen	MIC (mg/ml)	MBC (mg/ml)	
V. harveyi	0.10	0.61	
V. vulnificus	0.16	0.61	
V. parahaemolyticus	0.31	1.25	
Tetracyclines	0.03	0.03	
DMSO 30%	0.00	0.00	

Figure 4 Bioautogram of anti-V. harveyi (B), V. vulnificus (C) and V. parahaemolyticus (D).

(A) TLC chromatogram of the extract from S. hiroshimensis under uv light at 366 nm. (E) Reference antibiotic (tetracycline). The brilliant spots on chromatogram (B–E) are the inhibition zones. (H) TLC pattern under UV light at 254 nm. (F and G) TLC patterns sprayed by vanillin and anisaldehyde reagents. Photographed by Patchara Pedpradab, 1 April, 2024.

Media optimization

S. hiroshimensis (SK3 strain) was cultured in several media (Fig. 5). The results showed that the bacterium cultured in YM and ISP No. 3 showed the highest BSM production and hence was optimized for the future, as illustrated in Fig. 6. The results revealed that haft YM (YM/2) was the most appropriate medium when compared to other media (Fig. 6), as indicated by the potential of its anti-Vibrio bioactivity.

Figure 5 Effect of media types to BSMs production.

Data are shown as the value of mean ± SE to represent by bars. The difference letters mean statistically significant at 95% confidence interval (p < 0.05).

Figure 6 Media optimization to BSMs production.

Data are shown as the value of mean ± SE to represent by bars. The difference letters mean statistically significant at 95% confidence interval (p < 0.05).

Total protein production

Protein production in Streptomyces promote by various factors including media components and physical factors. Protein synthesis was stimulated by some trace elements and minerals contained in media, while the physical factor included pH and temperature as found in S. coelicolor and S. violaceruber (Giarrizzo, Bubis & Taddei, 2007). In case of S. coelicolor culture supplemented with wheat bran showed maximum protein releasing (4.5 mg/ml) in 6 144 h. Protein was released to the media generally appear during stationary phase and phase of decline due to nutrients starvation stress (Jayalakshmi et al., 2011; Mostafa et al., 2012). According to our result, it is revealed that the release of protein dramatically increased at 24 h, and the highest concentration (41.8 ± 6.36 mg/ml) remained at 96–144 h before decreasing, as illustrated in Fig. 7.

Figure 7 Protein secretion to liquid medium.

Effect of culturing media and other factors on the potential for bioactive secondary metabolite production

The optimum incubation temperature for BSM production in the SK3 strain was recorded at 30 °C (Fig. 8), as determined by the inhibition zone of 28.00 ± 0.00 mm. SK3 does not synthesize any BSMs at low temperatures (≤10 °C). The suitable incubation period was completed on the fifth and sixth days, as demonstrated by Fig. 9, wherein the inhibition zones of anti-V. parahaemolyticus, V. vulnificus, and V. harveyi were recorded at 15.00 ± 0.10, 37.00 ± 0.00, and 28.00 ± 0.05 mm, respectively. The proper agitation speed was recorded at 200 rpm with an inhibition zone of 35.00 ± 0.58 mm (Fig. 10), while the initial pH was observed at 7.00 with the inhibition zone at 33.00 ± 0.00 mm (Fig. 11). The suitable composition of carbon, nitrogen, and trace elements consisted of starch, malt extract, calcium carbonate (CaCO3), and magnesium sulfate (MgSO4), with inhibition zones of 23.00 ± 0.50 (Fig. 12), 23.33 ± 1.00 (Fig. 13), and 33.67 ± 0.58 and 33.67 ± 1.00 (Fig. 14), respectively. The proper salt contained in the media was 0.5%, determined by the inhibition zone at 32.67 ± 0.58 mm (Fig. 15).

Figure 8 Effect of initial incubation temperature to BSMs production.

Data are shown as the value of mean ± SE to represent by bars. The difference letters mean statistically significant at 95% confidence interval (p < 0.05).

Figure 9 Effect of incubation period to BSMs production.

Data are shown as the value of mean ± SE to represent by bars. The difference letters mean statistically significant at 95% confidence interval (p < 0.05).

Figure 10 Effect of agitation on BSMs production of S. hiroshimensis.

Data are shown as the value of mean ± SE to represent by bars. The difference letters mean statistically significant at 95% confidence interval (p < 0.05).

Figure 11 Effect of initial pH of medium on BSMs production of S. hiroshimensis.

Data are shown as the value of mean ± SE to represent by bars. The difference letters mean statistically significant at 95% confidence interval (p < 0.05).

Figure 12 Effect of carbon sources on BSMs production of S. hiroshimensis.

Data are shown as the value of mean ± SE to represent by bars. The difference letters mean statistically significant at 95% confidence interval (p < 0.05).

Figure 13 Effect of nitrogen sources on BSMs production of S. hiroshimensis.

Data are shown as the value of mean ± SE to represent by bars. The difference letters mean statistically significant at 95% confidence interval (p < 0.05)

Figure 14 Effect of trace elements on BSMs production of S. hiroshimensis.

Data are shown as the value of mean ± SE to represent by bars. The difference letters mean statistically significant at 95% confidence interval (p < 0.05).

Figure 15 Effect of salt concentration on BSMs production.

Data are shown as the value of mean ± SE to represent by bars. The difference letters mean statistically significant at 95% confidence interval (p < 0.05).

Primary chemical investigation of crude extract

Ethyl acetate extracts of S. hiroshimensis were primarily analyzed for the presence of BSMs by using TLC, Fourier transform infrared spectroscopy (FTIR), and proton nuclear magnetic resonance (1H-NMR) techniques. Each extract (1 mg/ml) was first applied on a TLC sheet, then developed in a mobile phase (chloroform and methanol 9:1, v/v), left to dry, and finally sprayed with vanillin, anisaldehyde, dragentdroff, and ninhydrin reagents. The compounds, upon reaction with vanillin and anisaldehyde reagents, mainly produced violet, green, pink, gray, and yellow colors, as shown in Fig. 4. The metabolites contained in the mixture were categorized into two main groups using LC chromatograms (Fig. 16). The first set of compounds absorbed UV light at 235, 254, and 285 nm, with peaks at retention times (Rt) of 1.443 and 6.589 min. The second set of compounds absorbed UV light at 366 nm, with Rt at 17.738 and 26.670 min. The other small peaks were identified as minor constituents. The 1H-NMR spectrum (Fig. 17) showed the presence of heteroatoms at pH 3.50–4.20 and an aromatic region at pH 6.40−7.10. Furthermore, the presence of methyl functions and aliphatic parts in the molecule was also observed between pH 0.90–2.60. This according to the FTIR spectrum (Fig. 18) revealed that the metabolites contained carboxylic acid functional groups at 2,925.17 and 1,711 cm−1 along with ester functional groups at 1,228.14 cm−1.

Figure 16 HPLC chromatograms of the extract of S. hiroshimensis.

Figure 17 1H-NMR of the crude extract of S. hiroshimensis.

Figure 18 FTIR spectrum of the crude extract from S. hiroshimensis.

Discussion

Streptomyces spp. are filamentous bacteria belonging to the class Actinimycetia that are widely distributed across both terrestrial and marine environments. It is well known that they produce potent antibiotic metabolites useful for pharmaceutical and agrochemical proposes. Marine Streptomyces are exposed to a critical and dynamic environment during their growth, such as, pressure, salinity, pH, and complex element forms. They have a high capacity for physiological adaptation by producing secondary metabolites, mainly antimicrobial compounds, in critical environments. Hence, they can be a potent source of new antibiotics. The SK3 strain was isolated from the Andaman Sea sediment in Thailand; its colony feature on YM agar showed an orange‒yellowish color with a circular shape and oily skin (Fig. 1). The cell morphology in the SEM micrograph (Fig. 1) showed a long, circular, cylindrical shape with branching aerial mycelium, indicating the characteristic Streptomyces features. The 16S rDNA sequencing revealed that the SK3 strain is similar to the previously described bacterium S. hiroshimensis, with a matching score of 99.40%; however, the phylogenic alignment showed branching out of S. hiroshimensis, as shown in Fig. 2. To clarify the species level, the biochemical characteristics were examined, and the results are shown in Table 1. The SK3 strain used glucose and inositol as carbon sources during the fermentation process. Additionally, it showed positive results for casein hydrolysis testing, indicating that it could degrade the casein protein by producing proteinase enzyme. This result was used to confirm that SK3 is similar to S. hiroshimensis BCRC1243, as previously reported by Chang et al. (2008). Physiological and morphological characteristics of marine Streptomyces vary across species, as they have been influenced by environmental parameters and geographical locations. Therefore, the diversity of these physiological values has been recorded. Generally, marine Streptomyces use glucose and fructose as the major carbon source with a salt requirement of approximately 0.5–3% and of up to 7% in halophytic species (Reddy, Ramakrishna & Rajagopal, 2011). The SK3 strain needed 0.5% sodium chloride (NaCl) and used glucose; this is similar to S. brasiliensis and Streptomyces spp. isolated from Yellow Sea sediment and the west coast of India (Lu et al., 2009; Remya & Vijayakumar, 2008) Streptomyces usually grow at temperatures of 25–37 °C, but some species grow at 20–45 °C depending on the environment and ocean zone (Fahmy, 2020). Streptomyces isolated from surface seawater survive over a wide range of temperatures (20–40 °C) at pH 4–10, but deep-sea species grow well at low temperatures and high salt concentrations (Kamjam et al., 2019; Risdian et al., 2021).

Selection of culture media

Fermentation optimization is needed to maximize production and low-cost investment. S. hiroshimensis was cultivated in nine kinds of media (ISP no. 2–7, MA, MHA and YM). The microbes cultured in YM and ISP No. 3 showed the highest potential for the synthesis of anti-vibriosis spp. metabolites (Fig. 5), and therefore, both of these media were optimized for use as the main media for S. hiroshimensis cultivation. The results showed that haft YM was the most suitable media (Fig. 5). The composition of nutrients is a critical growth factor and involves secondary metabolite synthesis in the Streptomyces cell. Generally, carbon and peptide sources play important roles in secondary metabolite production in Streptomyces (Hamed et al., 2018; Sánchez et al., 2010). The YM medium contains various carbon sources (yeast extract, malt extract, and dextrose sugar) and peptone as a protein source, whereas ISP No. 3 contains oatmeal, trace salt, and three kinds of electrolytes or minerals: ferric sulfate heptahydrate, manganese chloride tetrahydrate, and zinc sulfate hepahydrate. YM contains a higher concentration of carbon sources and protein than ISO No. 3, which has only oatmeal as a carbon and protein source; therefore, S. hiroshimensis prefers YM medium for growth and secondary metabolite production.

Media optimization for the production of BSMs

Primary metabolites and medium composition were found to be the precursors and essential elements for BSM synthesis in marine Streptomyces. In this study, the SK3 strain showed the best BSM production when cultured in haft YM (YM/2) medium compared with ISP No. 3 medium, as shown in Figs. 3 and 5. The YM medium contained yeast extract (3 g/litter), malt extract (3 g/litter), peptone and glucose in 5 and 10 g proportions, respectively. The ISP No. 3 medium is composed of oat meal (20 g/litter), ferric sulfate heptahydrate (0.001 g/litter), manganese chloride tetrahydrate (0.001 g/litter), and zinc sulfate heptahydrate (0.001 g/litter). It is assumed that the S. hiroshimensis SK3 strain prefers a carbon source for BSM synthesis rather than electrolyzing or trace minerals. This resembles the growth of the S. hiroshimensis BCRC12423 strain, which needs malt and yeast extracts as its main components and uses glucose as a carbon source (Chang et al., 2008). However, the full YM formula is not suitable for the production of BSMs when compared to haft YM because the production of secondary metabolites occurs when organisms grow in nutrient-starved environments (Fahmy, 2020).

Total protein production

Proteins regulate growth and stimulate antibiotic production in submerged cultures. The presence of extracellular proteins in the medium is a common phenomenon in Streptomyces culture because of biodegradation, carbon recycling, and other metabolic processes. The concentration of proteins in the medium implies the potential for the synthesis of secondary metabolites. Streptomyces secrete several heterologous polypeptides directly into the culture medium, thereby obviously benefitting the preservation of native conformation and antibiotic production (Anné et al., 2014; Hamed et al., 2018). Protein excretion generally takes place in the stationary phase, as found in S. coelocolor (Shahab et al., 1996). A few proteins in liquid culture have been discovered, and their functions have been defined. For example, EshA is needed for the growth of sporogenic hyphae and for the controlled synthesis of secondary metabolites, as observed in S. coelicolor and S. griseus (Čihák et al., 2017). The HyaS protein affects pellet morphology and conserves growth, whereas SsgA functions by stimulating fragmentation of hyphae (Koebsch et al., 2009; Sottorff et al., 2019; Van Wezel et al., 2000). The SK3 strain showed the highest total protein content on the fourth day (41.8 ± 6.36 mg/ml) and the fifth day (41.7 ± 0.03 mg/ml) after onset culture, followed by a gradual decrease (Fig. 7). Therefore, it may be assumed that the maximum protein was secreted during the stationary growth phase, wherein there was no growth of Streptomyces, followed by the dead phase caused by nutrient starvation.

Effect of media composition and other factors on the production of BSMs

The synthesis of BSMs in Streptomyces is influenced by the types and concentration of nutrients in the culturing media. Generally, the synthesis of BSMs arises during the late log phase as part of growth performance but varies according to media compositions and physical conditions (Sánchez et al., 2010). This includes sources of carbon and nitrogen, trace elements, salt content, pH, and agitation and incubation periods.

Effects of carbon, nitrogen, phosphorus and trace elements

All kinds of carbon sources can be utilized by Streptomyces, mainly to promote growth and often for the repression of secondary metabolite synthesis, for example, the reduction of actinomycin synthesis by S. antibioticus after the addition of glucose to the culturing media (Schrader & Blevins, 2001). Another case of repression of metabolites caused by the levels and types of carbon sources is the interference of S. Kanamyceticus in kanamycin production (Basak & Majumdar, 1973). This phenomenon is also observed in the SK3 strain, which uses glucose as a carbon source for growth but interferes with BSM synthesis, as illustrated by the inhibition zone in Fig. 12. Although nitrogen is an effective element for growth and secondary metabolite synthesis in Streptomyces spp., its effect varies according to organism strains and the types as well as concentrations of nitrogen sources. Several research studies have supported this theory. For example, S. naursei produced a 10-fold increase in the bioactive compound nystatin when it was submerged in media containing ammonium nitrate (Jonsbu, Ellingsenc & Nielsen, 2000). On the other hand, S. rimosus and S. griseocarneus need ammonium sulfate to synthesize oxytetracycline, enzyme protease, and isoquinoline antibiotics, respectively (AL-Ghazali & Omran, 2017; Rafique, Nawaz & Mukhtar, 2021; Yang & Lee, 2001).

S. hiroshimensis (SK3 strain) uses casein, peptone, beef extract, malt extract, and urea as nitrogen sources to produce BSMs. However, it shows the highest potency in antibiotics used against V. harvyi when unsupplemented by nitrogen, as shown in Fig. 13. This may result in complexity and variation in the pathways of secondary metabolism, which use the substances synthesized during primary metabolism processes as precursors for BSM production. Generally, marine Streptomyces produce and accumulate secondary metabolites when nutrient starvation occurs. In this case, the absence of nitrogen in the culture media may stimulate BSM synthesis. Streptomyces assimilates an optimum concentration of trace elements for growth and BSM synthesis; however, this assimilation varies according to the species and source trace elements. S. hiroshimensis (SK3 strain) requires carbonate, sulfate, and phosphate as trace elements for synthesizing anti-vibriosis spp. metabolites, as indicated by the inhibition zones at 33.67 ± 1.00, 33.67 ± 0.58, and 19.00 ± 0.58, respectively. However, the results of this study have revealed that mass production of BSMs does not require trace elements because antimicrobial activity remains at a high potential when unsupplemented by trace elements. CaCO3 does not indicate secondary metabolite production but stabilizes the pH fluctuation in a liquid medium. On the other hand, a few species, such as S. rimosus and S. albidoflavus, still need an optimum concentration of CaCO3 and MgSO4 for incorporation within the BSM synthesis process (Narayana & Vijayalakshmi, 2008; Schrader & Blevins, 2001). Some streptomyces do not favor MgSO4; for example, S. albidoflavas and S. tanashiensis (Singh, Mazumder & Bora, 2009). Phosphorus acts as a regulating factor for BSM synthesis via various biosynthesis pathways, depending on the type of Streptomyces, the sources of phosphorus, and their concentrations. In the case of the SK3 strain, dihyrodgen phosphate (H2PO4) does not exhibit a high potential for BSM production in comparison with CaCO3 and MgSO4 due to species variation (Yang & Lee, 2001).

Effect of initial pH

Generally, Streptomyces can grow and produce secondary metabolites at pH 7.0; however, this function differs according to the range of pH values (6.0–9.0) and the species. Changes in pH and extreme acidic or basic conditions can cause loss of growth and a decrease in BSM production due to a reduction in enzyme activity and interruption in proton pumping through the cell membrane (Yang & Lee, 2001). Our study found that the SK3 strain produced the most antibiotics at pH 7.0 and stopped the production of antibiotics at a pH lower than 5. Likewise, S. thermoviolaceus produced the highest concentration of an antibiotic compound, granaticin, at pH 6.5–7.5; on the other hand, it showed low production at a pH lower than 5.5 (James, Edwards & Dawson, 1991).

Effect of agitation speed

In the case of Streptomyces culture, the agitation speed regulates the growth rate, cell morphology, enzyme activity, and metabolic processes. Fast mixing (high agitation speed) results in increased growth and BSM production; however, too high a speed can induce cell damage, reduce growth, and decrease BSM synthesis (Zhou et al., 2018). Basically, the optimum agitation speed varies between 100–250 rpm, depending upon the species (AL-Ghazali & Omran, 2017). The SK3 strain showed the highest antibiotic production at an agitation speed of 200 rpm. Likewise, S. cuspidophorus showed the highest growth and antibiotic production when agitated at 200 rpm (Sholkamy et al., 2020).

Effect of sodium chloride concentration

NaCl involves osmotic pressure between the culture media and Streptomyces cells in a submerged culture system, while sodium ions (Na+) play an important role in membrane transport and maintaining ion equilibrium between intra- and extracellular fluid (Akond et al., 2016; Mansour, 2014; Gao et al., 2022; Kengpipat et al., 2016). Several Streptomyces species grow in the wild across a range of NaCl concentrations, depending on the environment and suitable NaCl values. Some Streptomyces have a deep-sea environment as their habitat; for example, S. bohaiensis can grow at a salt content of 11–37% (Imada et al., 2010; Pan et al., 2015). Increasing NaCl concentration causes growth reduction because both osmotic pressure and ionic equilibrium are interrupted; on the other hand, the reverse effect is that microbes accumulate high levels of secondary metabolites (Kengpipat et al., 2016). For instance, a streptomyces spp. that was cultured in high salt content (20 mg/l) produced a high concentration of actinomycin D (Singh, Mazumder & Bora, 2009). In this study, the SK3 strain was collected from an estuarine environment, which involved mixing of sea and river waters; therefore, it could survive across a wide range of salt concentrations. It showed high antibiotic production in media supplemented with 0.5% NaCl, determined by a maximum inhibition zone of 32.67 ± 0.58 mm. Likewise, S. cinereoruber isolated from an estuarine area produced high antibiotic concentrations in media supplemented with 3% NaCl (Praveen, Tripathi & Bihari, 2008).

Effect of incubation period

The incubation period affects secondary metabolite synthesis. Our study showed that BSM production started on the third day of the incubation period; the maximum BSM production was observed on the fifth day, and then the production gradually decreased. This pattern was also found in the Streptomyces R1 strain (Upadhyay et al., 2008; Kathiresan, Balagurunathan & Selvam, 2005). Kathiresan, Balagurunathan & Selvam (2005) reported that several Streptomyces strains isolated from terrestrial and marine environments showed the highest BSM production over 3–7 days, whereas a long incubation period (21 days) was recorded for Streptomyces SA404 (Bundale et al., 2015). It should be noted that the incubation period can vary as per the species and in relation to environmental conditions. Secondary metabolites are generally synthesized during the stationary growth phase, where no significant growth is recorded due to nutrient starvation and some toxic metabolites are released into the cultured media. This situation triggers a biochemical signal for secondary metabolite synthesis and BSM accumulation in cells (Singh, Jain & Shri, 2021). Therefore, an optimal incubation period might be determined based on the potential of BSM activity against tested pathogens.

Biological screening and primary chemical analysis

The excessive use of antimicrobial drugs as prophylaxis in aquaculture water has led to the emergence of AMR. Vibriosis is one of the most serious bacterial diseases affecting marine shrimp culture farms. Several antibiotics have been applied to control vibriosis outbreaks worldwide, particularly in Southeast Asian regions. The discovery of new antibiotics for vibriosis treatment is important and urgent. Streptomyces are a rich source of such antibiotics. In this experimental study, the crude extract of the Streptomyces SK3 strain showed antimicrobial activity against the vibriosis pathogens V. parahaemolyticus, V. vulnificus, and V. harveyi (Fig. 3). Meanwhile, the MIC and MBC values indicated that the extract showed the highest potency against V. harveyi (Table 3). Other researchers reported that compounds such as ganamycin A and D isolated from S. rosa var noloensis showed antimicrobial properties against V. parahaemolyticus K-1 with MICs of 3.10 and 0.05 μg/ml; furthermore, the compound chaxalactin isolated from Streptomyces has MIC values of 12.5–20 μg/ml, and the benzaldehyde derivative active against V. harveyi has an MIC of 20 μg/ml (Cho & Kim, 2012; Hayashi et al., 1982; Rateb et al., 2011). Our MIC showed higher values than the former reports because it was a crude extract; in the case of pure compounds, the MIC should be at a lower concentration. Diverse routine techniques, such as TLC, HPLC and LC‒MS/MS, have been primarily used to investigate the metabolites contained in natural extracts. For TLC analysis, several spray reagents were applied on the TLC sheet to identify the types of compounds. Vanillin was used to detect terpenoids, steroids, phenolic compounds, and aromatic amines, while anisaldehyde was typically used to detect steroids, terpenoids, and some fatty acids (Fathoni et al., 2020; Maya et al., 2019). In this experimental study, the Streptomyces compounds reacted with vanillin to produce violet, green, yellow, and gray bands (Fig. 4), thereby indicating the presence of steroids, terpenoids, and some reducing carboxylic acid compounds. According to TLC performed after spraying with the anisaldehyde reagent, pink and orange bands indicated the presence of steroids and terpenoids, whereas yellow bands indicated the presence of fatty acids. Based on chromatograms derived from liquid chromatography (LC) (Fig. 16), the compounds absorbed UV light at 235, 254, and 285 nm to carry on double bonds in molecules (or unsaturated compounds). On the other hand, compounds absorb UV at 366 nm to form saturated molecules. Compounds with BSM molecules contained carboxylic acid groups, as interpreted by the presence of infrared (IR) bands at 2,925.17 and 1,711 cm−1. 1H-NMR signals resonating at pH 3.50–4.20 were used to confirm that the BSMs carried heteroatoms within molecules; also, the percentage of aromatic fragment was identified by the signals at pH 6.40–7.10.

Conclusions

The long period needed and the misuse or overuse of artificial antibiotics during shrimp farming results in drug resistance phenomena. The discovery of natural antibiotics constitutes an important means to enhance prototype molecules for the development of new, effective, and eco-friendly drugs. In this study, a marine actinobacteria, S. hiroshimensis, was found to produce highly potent BSMs against several vibriosis pathogens infecting white shrimp (Litopenaeus vannamei), including V. harveyi, V. parahaemolyticus, and V. vunificus. They produced the highest BSMs when cultured in YM/2 medium at pH 7 with an agitation speed of 200 rpm. It was not necessary to supplement carbon, nitrogen, and any trace elements during the culture process to produce BSMs, but 0.5% salt was added. Secondary metabolites produced by S. hiroshimensis included terpenoids, steroids, and fatty acids. The compounds carried on the carboxylic acid and ester functions within the molecule. Based on the results of this study, the extract obtained from S. hiroshimensis may be used as a natural antibiotic product against vibriosis pathogens in shrimp culture farming.

Supplemental Information

Supplemental Information 1 Supplementary data.

Supplemental Information 2 Raw data condition.

Supplemental Information 3 Data used for the preparation of Figure 10 (agitation speed).

Raw data exported from the statistical software SPSS (version 22) was analyzed using one-way ANOVA at a 95% confidence interval (p < 0.05) of agitation speed.

Supplemental Information 4 Data used for the preparation of Figure 12 (carbon sources).

Raw data exported from the statistical software SPSS (version 22) was analyzed using one-way ANOVA at a 95% confidence interval (p < 0.05) of carbon sources.

Supplemental Information 5 Data used for the preparation of Figure 7 (protein secretion to liquid medium).

Raw data exported from the statistical software SPSS (version 22) was analyzed using one-way ANOVA at a 95% confidence interval (p < 0.05) of protein secretion to liquid medium.

Supplemental Information 6 Data used for the preparation of Figure 9 (incubation period).

Raw data exported from the statistical software SPSS (version 22) was analyzed using one-way ANOVA at a 95% confidence interval (p < 0.05) of incubation period.

Supplemental Information 7 Data used for the preparation of Figure 9 (agitation speed).

Raw data exported from the statistical software SPSS (version 22) was analyzed using one-way ANOVA at a 95% confidence interval (p < 0.05).

Supplemental Information 8 Data used for the preparation of Figure 13 (nitrogen sources).

Raw data exported from the statistical software SPSS (version 22) was analyzed using one-way ANOVA at a 95% confidence interval (p < 0.05) of nitrogen sources.

Supplemental Information 9 Data used for the preparation of Figure 11.

Raw data exported from the statistical software SPSS (version 22) was analyzed using one-way ANOVA at a 95% confidence interval (p < 0.05) of pH.

Supplemental Information 10 Data used for the preparation of Figure 6.

Raw data exported from the statistical software SPSS (version 22) was analyzed using one-way ANOVA at a 95% confidence interval (p < 0.05) of medium formula.

Supplemental Information 11 Data used for the preparation of Figure 15.

Raw data exported from the statistical software SPSS (version 22) was analyzed using one-way ANOVA at a 95% confidence interval (p < 0.05) of salt concentration.

Supplemental Information 12 Data used for the preparation of Figure 8.

Raw data exported from the statistical software SPSS (version 22) was analyzed using one-way ANOVA at a 95% confidence interval (p < 0.05) of incubation temperature.

Supplemental Information 13 Data used for the preparation of Figure 7.

Raw data exported from the statistical software SPSS (version 22) was analyzed using one-way ANOVA at a 95% confidence interval (p < 0.05) of protein secretion to liquid medium.

Supplemental Information 14 Data used for the preparation of Figure 14.

Raw data exported from the statistical software SPSS (version 22) was analyzed using one-way ANOVA at a 95% confidence interval (p < 0.05) of trace elements.

Supplemental Information 15 Data used for the preparation of Figure 5.

Raw data exported from the statistical software SPSS (version 22) was analyzed using one-way ANOVA at a 95% confidence interval (p < 0.05) of type of medium.

Additional Information and Declarations

Competing Interests

Author Contributions

Data Availability

The authors declare that they have no competing interests.

Rachow Khaochamnan conceived and designed the experiments, performed the experiments, analyzed the data, prepared figures and/or tables, and approved the final draft.

Naraid Suanyuk performed the experiments, prepared figures and/or tables, and approved the final draft.

Monthon Lertcanawanichakul performed the experiments, analyzed the data, prepared figures and/or tables, and approved the final draft.

Patchara Pedpradab conceived and designed the experiments, performed the experiments, analyzed the data, prepared figures and/or tables, authored or reviewed drafts of the article, and approved the final draft.

The following information was supplied regarding data availability:

The raw data is available in the Supplemental File.

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
