# Peer review of "Biological characteristics of marine Streptomyces SK3 and optimization of cultivation conditions for production of compounds against Vibiriosis pathogen isolated from cultured white shrimp (Litopenaeus vannamei)"

_PeerJ, doi:10.7717/peerj.18053_

## Round 0.1 · original submission · Major Revisions

· Academic Editor

Major Revisions

Two reviewers have reviewed the manuscript and suggested significant changes. Please spend extra care in describing the methods you used.

**Language Note:** The review process has identified that the English language must be improved. PeerJ can provide language editing services - please contact us at [email protected] for pricing (be sure to provide your manuscript number and title). Alternatively, you should make your own arrangements to improve the language quality and provide details in your response letter. – PeerJ Staff

Reviewer 1 ·

Basic reporting

1- A lot of redundant sections. Comments are provided in the comments file.
2- There is a need to add work flow diagram of something like that cause the text alone is difficult to follow.

Experimental design

The authors tried to carry out a comprehensive study on isolation, characterization and optimization marine Streptomyces SK3 for producing anti-Vibriosis. It seems the authors did many small experiments and combined them into 1 article. Unfortunately, if the goal is optimization, this approach has too many disadvantages, which were apparent in this manuscript as the work has major flows that are summarized below.

There is also no use of statistical analysis.

Many experiments in the methodology section appear to not be standard (e.g. CLSI) and no citations were provided (e.g. MIC was carried out according to xyz 2017)

Other comments in the comments file.

Validity of the findings

Some experimental data not valid as its not standard method.

No statistical analysis.

The isolation part is valid.

Several experiments were done with inconstant intervals. Which is not what we need when we want to carry out optimization experiments.

Additional comments

Title:
The title includes “implications for reducing antibiotic resistance in an aquaculture environment” which is not within the scope of the experiments done and presented in the manuscript. Maybe if there was some experiment on synergistic activity between the Streptomyces extract and commercial antibiotics that would be fine. Or there was study on the expression of AMR genes by the studied vibrio spp.

Abstract:
• Please include the compound possible structure in the concluding sentence of the abstract

Introduction
• Line 82-86, please elaborate a bit on why the alternative metabolites should be from natural resources.
• Line 87-111, there is a lot of overlapping statements. Several points are mentioned multiple times. Please rearrange and shorten the text, its very wordy.
• Line 135-149, includes streptomycetes as a source of novel antibacterial compounds, culture conditions that affect the production of BSM, etc. These points have already been mentioned in L117-119 and 110-112, L95-97, etc. Please rearrange the text and combine similar sections.
Materials and methods
• A general comment is the flow of optimization is confusing. Its important to screen factors in 1 experiment because often factors interact. For example, media type, temperature and carbon source may have significant interaction. Another example is lets say YM give the best results initially when tested at 30C, but if ISP3 is cultured at 40C it might give even better results.
• Another general comment is the intervals used in several experiments is confusing. MIC (L265-276) stated extract range to be 10-0.15 mg/ml, however, table 3 contains 0.101±0.02, which is not in the mentioned range.
• Other troublesome intervals are also observed in figure 10 (agitation speed) where the range is 0, 16, 20, 35, and back to 23. Also in figure 15 that shows salt starting with an interval of 0.1, then jumping to 0.5, and then 1.1 (2.5 to 3.6).
• Line 222-229, the names of media mentioned here and in Figure 5 are not the same. Aren't all those media suppose to be in broth form?
• Line 228 and 229, can you please elaborate on “potency” and “high growth”? What is the cutoff you considered “potent”? For example, >15 mm.
• Line 231-237, why didn’t the author use a systematic approach such as a response surface method (RSM) to optimize the culture conditions? Because there are so many dependent and independent variables.
• Line 249-252, please provide the isolation process and the accession numbers of these isolates to have a complete and transparent method. Maybe the isolates were provided by another lab or have been published elsewhere. Just include this information.
• Line 255-263, No citation. The authors used 30 C for all V. harveyi, V. vulnificus and V. parahaemolyticus which is not ideal. Literature shows the optimum temperatures for those species are 26, 36 and 37.
• Line 265-276, No citation. The MIC and MBC appear to be done not according to CLSI method. Please clarify. The starting inoculum size wasn’t provided. It could be 108 or 105, not clear. The use of 690 nm is also not justified as the normal ranges used to measure bacterial growth is 590-630 nm. Also using MHA in the MBC could be troublesome as marine bacteria need salt. CLSI requires the use of MHA2, which is MHA supplemented with 1-2% salt.
Results
• A general comment is the lack of statistical analysis throughout the manuscript. Data must be analyzed statistically to show the significant differences in all the results presented.
• Table 3, the MIC is shown as mean +/- standard deviation/error. This is strange as Mic usually its done in 2 or 3 fold with replicates. Meaning no standard deviation or standard error is reported.
• Line 310, mentioned here “extract and supernatant”. I didn’t find “supernatant anywhere in the text before this line which is troublesome for me.
• Lin 360, “pH 0.90-2.60” not clear what this means. Is it pH or retention time?
• Figure 16 needs a caption to explain the 4 pictures.
Discussion
• Line 417-430, the authors did attempt to discuss the optimization process, however, considering the huge number of factors involved, in which the author studied several of them, this text is too shallow and need much more elaboration.
• Line 433-450, did the author consider bacteriocins as “active protein” that might be the main active ingredient here?
• Line 452-457, again its really insufficient to discuss the effect of more than 5 factors (carbon and nitrogen, trace elements, salt content, and agitation and incubation periods) within 5 lines of text. Should combine up to line 499.
• Line 510-517, states that “highest antibiotic production at 200 rpm” However, figure 10 shows that the highest inhibition zone was at 350 rpm.
• L570-585, I think the authors discuss the results maybe spot (compound) by spot. There are many spots being discussed in terms of many assays and its hard to follow. Moreover, all the analysis done seems to not have helped in pinpointing what the active compound is. The author is still left with large number of possibilities being steroids, terpenoids, reducing carboxylic acid, fatty acids.

·

Basic reporting

The MS entitled “Characterization and cultural optimization of a marine Streptomyces SK3 for producing anti-Vibriosis pathogens infecting the cultured white shrimp (Litopenaeus vannamei): implications for reducing antibiotic resistance in an aquaculture environment (#94971)” is an extensive work done by Rachow Kaochamnan et al. The MS is interesting study related to the characterization of Streptomyces SK3 to produce antimicrobial secondary metabolites. The MS need major revision before publication

Specific Comments
1. Title is vague. Reframe the title
2. Streptomyces belongs to Actinomycetes. Unfortunately the authors used the term bacteria, the entire MS. Please change to “Actinomycetes”
3. 239: The extraction protocol need elaborate
4. Line 325: This section need elaboration with literature
5. Need production and optimization study for the secondary metabolite production with statistical methods by using RSM (Response surface methodology)
6. The MS need updating the entire MS especially way of writing and rectify typographical errors
7. No statistics analysis done. Analyse the data with post ANOVA

Experimental design

1. 239: The extraction protocol need elaborate
2. Line 325: This section need elaboration with literature
3. Need production and optimization study for the secondary metabolite production with statistical methods by using RSM (Response surface methodology)

Validity of the findings

Findings is good

Additional comments

Nil

---

## Round 0.2 · Minor Revisions

· Academic Editor

Minor Revisions

Please address the questions relating to the Response Surface Methodology and another minor issue with figure 4

·

Basic reporting

The MS entitled “Biological characteristics of marine Streptomyces SK3 and optimization of cultivation conditions for production of compounds against Vibiriosis pathogen isolated from cultured white shrimp (Litopenaeus vannamei) (#94971)” is an interesting study about the influence of the secondary metabolites able to control vibrio pathogen in shrimp aquaculture industry. The MS is well-written and organized. Streptomyces yielding secondary metabolites had several advantages effectively controlling the bacterial and viral pathogens in the aquaculture operations were reported already. The MS need a few improvements before publication
Specific comments
1. Labelling need in Figure 4, bioautogram
2. Why can’t the Response Surface Methodology (RSM) for production and optimization. Please include the RSM protocol

Experimental design

Why can’t the Response Surface Methodology (RSM) for production and optimization. Please include the RSM protocol

Validity of the findings

Good

Additional comments

No

---

## Round 0.3 · accepted · Accept

· Academic Editor

Accept

The authors have done a good job addressing all points of the reviewers.